# Study of Reaction Parameters for the Precise Synthesis of Low-Molecular-Weight Oligosiloxanes

**DOI:** 10.3390/ma18245677

**Published:** 2025-12-17

**Authors:** Satoru Saotome, Jiaorong Kuang, Yujia Liu, Takayuki Iijima, Masafumi Unno

**Affiliations:** 1Department of Chemistry and Chemical Biology, Gunma University, 1-5-1 Tenjin-cho, Kiryu 376-8515, Japan; satoru.saotome.ma@mcgc.com (S.S.); t242a002@gunma-u.ac.jp (J.K.); 2Polyester Laboratory, Polymers Research & Development Center, Mitsubishi Chemical Corporation, 1, Toho-cho, Yokkaichi-shi 510-8530, Japan; takayuki.iijima.ma@mcgc.com

**Keywords:** oligosiloxane, ring-opening polymerization, molecular weight control, degree of polymerization, controlled synthesis

## Abstract

This study investigates the influence of various parameters on the synthesis of oligosiloxanes with degrees of polymerization below 15. The work provides insights into methods for synthesizing oligosiloxanes with precisely controlled molecular weight and degrees of polymerization. Low-molecular-weight polysiloxanes with well-defined molecular characteristics have attracted attention due to their versatile functional properties and potential applications. Although some studies have explored the control of polysiloxane molecular weights, precise regulation of oligosiloxane molecular weight has been rarely investigated. This study aims to establish optimized reaction conditions for the synthesis of oligosiloxanes with precisely controlled molecular weights. The results reveal that the molecular weight of oligosiloxanes can be effectively tuned by adjusting the molar ratio between the promoter and initiator, the initiator and cyclotrisiloxane (D_3_), as well as by varying the lithium type and solvent composition in the ring-opening polymerization of D_3_. These findings provide valuable guidance for tailoring oligosiloxane properties and expanding their potential applications in advanced materials.

## 1. Introduction

Siloxanes are compounds based on silicon–oxygen bonds as their fundamental backbone. Among these, the linear structure siloxanes composed of consecutive D units (silicon atoms bonded to two oxygen atoms) are called polysiloxanes or oligosiloxanes. They have long attracted attention due to their simple structure and unique properties. For example, polysiloxanes or oligosiloxanes are known for their heat resistance, chemical stability, water repellency, impact resistance, gas permeability, excellent dielectric characteristics, and superior optical properties [1,2,3]. Taking advantage of these properties, they have been applied in various fields such as resin modifiers [4,5], comonomers [6,7], gas–liquid separation membranes [8], explosives [9], sensor systems [10], and pharmaceutical applications [3].

Polysiloxanes are generally synthesized either by the hydrolysis/dehydration of chlorosilanes or alkoxysilanes [11,12], or by the ring-opening polymerization of cyclic siloxanes catalyzed by acids or organobases [13,14,15,16,17,18,19]. However, while the former approaches for accurately controlling the molecular weight of polysiloxanes remain limited, the latter method—though generally considered to yield narrow molecular weight distributions—often produces polysiloxanes with a certain degree of molecular weight variation. Although this variation may be negligible for polysiloxanes with degrees of polymerization exceeding 50, it becomes a significant source of error for oligosiloxanes with degrees of polymerization below 15, potentially leading to considerable differences in their physical properties, which are strongly dependent on molecular weight [20,21]. Therefore, synthetic methods that enable precise control over molecular weight are highly needed. However, precise molecular weight control of polysiloxanes or oligosiloxanes using the above-mentioned methods has not been previously reported.

Recently, Matsumoto et al. reported methods involving sequential B(C_6_F_5_)_3_-catalyzed dehydrocarbonative cross-coupling and hydrosilylation that enable ultra-precise control not only over molecular weight but also over sequence [22,23,24,25]. They later simplified this approach to a one-step, B(C_6_F_5_)_3_-catalyzed one-pot process for preparing well-defined oligosiloxanes [26]. Although these methods provide exceptional precision, even for oligosiloxanes, they require multi-step operations and rely on an expensive borane catalyst, which limits their practicality for industrial-scale production.

In addition, there have been reports on the use of guanidines [15,27] and imidazoles [28] as specialized organic bases, as well as methods employing a combination of water and strong bases [29], or utilizing photochemical reactions [30]. However, challenges such as the high degrees of polymerization of the target polysiloxanes and the complexity of the operational procedures still remain.

Given this background, this study aims to systematically explore anionic ring-opening polymerization conditions of cyclotrisiloxane that enable precise molecular weight control of oligosiloxanes with degrees of polymerization below 15.

## 2. Materials and Methods

### 2.1. Equipment

The Fourier transform nuclear magnetic resonance (NMR) spectra of oligosiloxanes in this study were measured using a JNM-ECA 400 (JEOL, Tokyo, Japan) (^1^H at 399.78 MHz) NMR instrument manufactured by JEOL. For ^1^H NMR, chemical shifts were assigned as δ units (ppm) relative to tetramethylsilane (TMS), and the residual solvent peaks were used as standards. The spectra were acquired with complete proton decoupling. Matrix-assisted laser desorption/ionization coupled time-of-flight (MALDI-TOF) mass was analyzed by a Shimadzu (Kyoto, Japan) AXIMA Performance instrument, using 2,5-dihydroxybenzoic acid (dithranol) as the matrix and AgNO_3_ as the ion source. All reagents used in the analysis were of analytical grade.

### 2.2. General Consideration of Synthesis

The oligosiloxanes were synthesized under an argon atmosphere using standard Schlenk techniques, unless otherwise noted. Tetrahydrofuran (THF) and toluene were used after being purified with an mBRAUN solvent purification system. Trimethyltrivinylcyclotrisiloxane, hexamethylcyclotrisiloxane, and chlorodimethylsilane were purchased from Tokyo Chemical Industry Co., Ltd. (Tokyo, Japan). Toluene, diethyl ether, hexane and Na_2_SO_4_ were purchased from FUJIFILM Wako Pure Chemical Corporation (Tokyo, Japan). *n*-Butyllithium (*n*-BuLi, 1.6 M in hexane) and methyllithium (MeLi, 1.04 M in Et_2_O) were purchased from Kanto Chemical Co., INC. (Tokyo, Japan). All reagents were used as received without further purification.

### 2.3. Optimization of Reaction Time for Initiator Formation



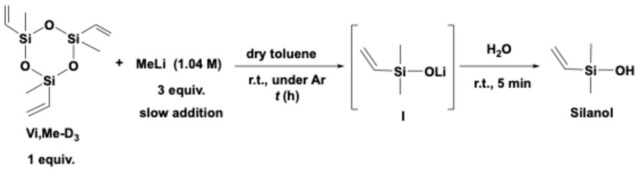



An argon-purged Schlenk flask equipped with a stir bar was charged with dry toluene (0.54 mL) and trimethyltrivinylcyclotrisiloxane (**Vi,Me-D_3_**) (0.54 mL, 2 mmol). The solution was cooled to 0 °C and methyllithium was added dropwise as 1.04 M solution in Et_2_O (5.8 mL, 6.0 mmol) into the reaction mixture under argon atmosphere at 0 °C. After the addition, the reaction mixture was stirred at room temperature for 30 min, 1 h and 2 h, respectively. The reaction mixture was then quenched with water (5.8 mL), and the mixture was stirred at room temperature for 5 min to convert the Si-OLi species to the corresponding Si-OH silanol. The mixture was extracted with diethyl ether, and the combined organic phases were washed three times with brine. The combined organic layers were dried over anhydrous Na_2_SO_4_ and concentrated by rotary evaporation at room temperature to afford a crude mixture as a transparent colorless liquid, which was used immediately to record its ^1^H NMR spectrum without isolation.

### 2.4. Typical Procedure for the Synthesis of Oligosiloxane (e.g., Table 3, Entry 3)



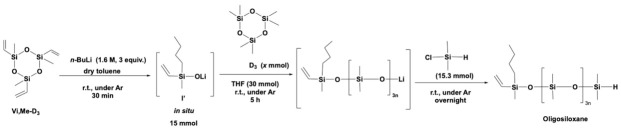



An argon-purged, three-necked, 100 mL round-bottom flask fitted with a stir bar was charged with trimethyltrivinylcyclotrisiloxane (**Vi,Me-D_3_**) (1.35 mL, 5.0 mmol) and dry toluene (1.50 mL). The solution was cooled to 0 °C, and *n*-BuLi was added dropwise as 1.6 M solution in hexane (9.40 mL, 15.0 mmol) into the reaction mixture under argon at 0 °C. The mixture was stirred for 30 min at 0 °C, after which hexamethylcyclotrisiloxane (D_3_) (13.4 g, 60.0 mmol) was introduced, followed by dry THF (2.50 mL, 30.8 mmol) as a polymerization promoter at room temperature. Stirring was continued for 5 h at room temperature, and polymerization was terminated by addition of a slight excess of chlorodimethylsilane (1.67 mL, 15.3 mmol). The solution was stirred overnight at room temperature, extracted three times with hexane, and combined organic layers were washed three times with brine. The solvent was removed by a rotary evaporator for 1.5 h at 35 °C, and the residue was dried under vacuum at 60 °C for more than 3 h to yield oligosiloxanes (9.60 g, average Si number = 12.2, n = 3.4) as a transparent colorless liquid.

The MALDI-TOF MS spectrum of the obtained oligosiloxane is shown in Figure 1. Prominent peaks corresponding to Si numbers of 14, 17, and 20 (i.e., n = 4, 5, and 6) were observed. The peak at *m*/*z* = 242.21 corresponds to the compound with two silicon at-oms (n = 0), and this species is considered to contribute to the lower average silicon number calculated from the 1H NMR data compared to that of the main peaks observed in the MALDI-TOF MS spectrum.

The other syntheses of oligosiloxanes described in this paper were also performed using the same procedure, with only the species and amounts of the reagents varied.

Argon can be substituted with nitrogen, which is widely used in industry and economically feasible.

## 3. Results and Discussion

### 3.1. Optimization of Reaction Time for Initiator Formation

The reaction conditions for the formation of lithium silanolate were first optimized. Lithium silanolate functions as an initiator for the ring-opening polymerization of 1,3,5-trimethyl-1,3,5-trivinylcyclotrisiloxane (**Vi,Me-D_3_**).

To generate lithium silanolate (I), three equivalents of methyllithium (MeLi) were slowly added to a solution of vinylmethylsiloxane in dry toluene at 0 °C, and the mixture was stirred for a controlled period at room temperature. The reaction was subsequently quenched with water (Figure 1). Figure 2 shows the ^1^H NMR spectra obtained after reaction times of 0.5, 1, and 2 h. The integration ratio of the methyl peak near 0 ppm to the vinyl peaks around 6 ppm for silanol was 6:3 at the 0.5 h mark, indicating that the reaction had already reached completion by that time.

Based on these results, the conditions corresponding to Entry 1 (Table 1) were selected as optimal. In all subsequent experiments, the lithium silanolate formation was carried out by stirring at room temperature for 30 min.

### 3.2. Investigation of Reaction Conditions for Propagation of Ring-Opening Polymerization

Next, the siloxane chain propagation process of the ring-opening polymerization (ROP) was optimized. Two parameters were examined: reaction time and promoter concentration. In this study, the ratio of hexamethylcyclotrisiloxane (D_3_) to lithium siloxane initiator (I) was set to 0.22, following the procedure reported in the literature for the preparation of polysiloxanes [14]. In that work, the initiator (I) was prepared from a pre-prepared methyllithium (MeLi) solution in hexane, whereas in this study, we prepared initiator (I) from commercially available MeLi solution in diethyl ether (Et_2_O). The influence of the solvents will be discussed in Section 2.4.

The chain length of the oligosiloxanes was evaluated based on the average number of silicon (Si) atoms contained within the formed oligosiloxanes and the disiloxane obtained from excess initiator and chlorosilane (Figure 2). This average value was determined by the ^1^H NMR integration of methyl groups on Si atoms, represented as “Average Si number” in Table 2, and can be interpreted as the degree of polymerization (DP) of the oligosiloxanes.

When the amount of promotor THF was fixed at 2 or 0.2 equivalents related to initiator (i.e., ([THF]/[I] = 2 or 0.2), the average number of Si atoms decreased slightly with increasing reaction time from 5 h to 24 h (Table 2, Entries 1 and 2; Entries 3 and 4). This decrease is likely due to a backbiting process occurring during prolonged reaction times, which is consistent with the classical kinetic behavior reported for the ROP of D_3_ [31,32].

It has been reported that the DP of polysiloxanes obtained via the ROP of D_3_ can be controlled by reducing the ratio of promoter to initiator [33]. Lowering the promoter-to-initiator ratio can slow down the propagation rate while simultaneously suppressing backbiting reactions. To examine whether the DP can also be controlled by the promoter-to-initiator ratio in the synthesis of oligosiloxanes with DPs of 15 or less, we reduced the amount of promotor THF by a factor of 10 while keeping the reaction time constant. The DP of the resulting oligosiloxanes remained nearly unchanged (Table 2, Entries 1 and 3; Entries 2 and 4). These results indicate that, in the synthesis of oligosiloxanes with DPs not exceeding 15, the promoter concentration has only a minor influence on the final oligomer chain length.

### 3.3. Variation in Molecular Ratio of Cyclotrisiloxane

To achieve precise control over the degree of polymerization (DP) of the oligomers, we hypothesized that stoichiometric regulation of D_3_ would be essential. By increasing the stoichiometric ratio of D_3_, longer oligosiloxanes could be obtained. Therefore, the experiments were conducted by varying only the molar ratio between D_3_ (hexamethylcyclotrisiloxane) and the initiator, while keeping the reaction time (5 h) and promotor concentration ([THF]/[I] = 2) constant. The lithium source used in this part to generate the corresponding initiator is commercially available *n*-butyllithium (*n*-BuLi) in hexane solution, which is less expensive and more accessible.

In the stoichiometric reaction, the lithium silanolate initiator attacks the Si-O bond of a D_3_ molecule to open the cyclic structure, after which polymerization proceeds with D_3_ in the presence of the promotor THF, followed by end-capping with monochlorosilane (Figure 3). If the reaction is precisely controlled and D_3_ is completely consumed, the theoretical number of Si atoms in the obtained oligosiloxanes can be calculated using the following equation:$$\mathrm{Si}\;\mathrm{number}=\;3n+\;2,\;n=\frac{[D3]mol}{[I]mol}$$

The obtained results indicate that increasing the molar ratio of D_3_ relative to the initiator (I) led to a corresponding increase in the Si numbers (i.e., the number of Si atoms) of the resulting oligosiloxanes, reflecting an increase in their DP. However, the Si numbers estimated from the ^1^H NMR integration of methyl groups on Si atoms were smaller than the calculated values (Table 3), which can be attributed to the presence of unreacted D_3_ remaining in the reaction system or D_3_ generated via backbiting process. Due to its high volatility, D_3_ is considered to be removed from the oligosiloxanes during purification and vacuum drying processes. Therefore, the amount of D_3_ in the final product is difficult to detect and quantify.

Based on this observation, we hypothesized that if D_3_ were completely consumed, the degree of polymerization of the oligosiloxanes would reach the theoretical value. To test this hypothesis, we attempted to increase the solubility and conversion of D_3_ by modifying the composition of the reaction solvents.

### 3.4. Study of the Solvent Composition for Ring-Opening Polymerization

In this section, hexane that appears in the reaction generally originates from the *n*-BuLi solution, while toluene is introduced during the formation of the lithium silanolate initiator (I′), without any further addition. Only in Experiments 4 and 6 (Table 4) were additional hexane and toluene added together with the promoter, THF. Hexane and toluene are regarded as nonpolar solvents, whereas THF is considered an aprotic polar solvent. As the amount of THF decreased, the THF/I′ ratio decreased and, consequently, the ratio of nonpolar solvent to aprotic polar solvent (Np/P) increased. Correspondingly, the average number of Si atoms in the formed oligosiloxanes decreased (Table 4, Entries 1–4; Entry 5 and 6). When D_3_/I′ = 1.5, THF/I′ = 2.1 and Np/P = 3.20, the estimated Si number was closest to the calculated value (Table 4, Entry 4). These results clearly demonstrate that an increase in the Np/P ratio (i.e., a decrease in the THF/I ratio) leads to a marked decrease in polymer chain length. As shown in Figure 3, the size of the circles increases along the positive direction of the *y*-axis, indicating that increasing the molecular ratio of THF relative to the initiator leads to longer oligosiloxane chains. Conversely, the circle size decreases along the positive direction of the *x*-axis, suggesting that a higher proportion of nonpolar solvent in the reaction system results in oligosiloxanes with lower molecular weights.

Aprotic polar solvents are known to coordinate with the lithium silanolate active end groups, thereby stabilizing them and suppressing their quenching. The observed increase in the degree of polymerization (DP) with decreasing Np/P ratio is considered to result from this stabilizing effect of the aprotic polar solvent on the lithium silanolate termini.

It is indeed necessary to take the influence of concentration into account in ROP. However, in the present experiments, no clear correlation was observed between the concentration and the degree of polymerization (DP) of the obtained oligosiloxanes.

In the case of using methyllithium (MeLi) to produce silanolate initiator, instead of hexane, diethyl ether appeared as nonpolar solvent in the reaction originating from the MeLi solution. However, unlike *n*-BuLi, the significant difference in the DP of the formed oligomers has not been observed varying Np/P, which has little effect on DP.

This difference between MeLi and *n*-BuLi would be attributed to the different steric bulk of methyl and *n*-butyl groups. When MeLi is used as the lithium source, its smaller steric bulk compared to *n*-BuLi is expected to enhance the reactivity of the active end groups. Since the formation of oligosiloxanes involves polymer chains with a DP around 10, the reaction is particularly sensitive to steric effects at the opposite terminus of growing chain ends. Therefore, the difference in steric hindrance between methyl and butyl end groups likely leads to differences in the reactivity of the lithium silanolate termini. As a result, oligosiloxanes with methyl end groups, which are more reactive, may be less responsive to DP control via the Np/P ratio. This difference in reactivity may also be attributed to the solvation of lithium cations, since diethyl ether (Et_2_O) is used instead of hexane in the MeLi system. A more detailed investigation is required to substantiate this mechanistic assumption.

## 4. Conclusions

In conclusion, this study demonstrated that by optimizing the reaction conditions, the molecular weight of oligosiloxanes with degree of polymerization (DP) below 15 can be precisely controlled. In the ring-opening polymerization of D_3_, by adjusting the molecular ratio between the promoter and initiator, the initiator and D_3_, as well as by varying the lithium silanolate type, the DP of the resulting oligosiloxanes could be effectively tuned. Furthermore, modifying the solvent composition used in the reaction also enabled fine control over the DP to achieve the desired values.

By applying the findings of this study, oligosiloxanes with precisely controlled DP can be synthesized. Such oligosiloxanes are expected to exhibit higher functionality compared to siloxanes with uncontrolled polymerization. Their potential applications include use as precision materials and functional monomers [6]. This research provides fundamental insights that significantly contribute to the material development of siloxane-based compounds.

## Data Availability

The original contributions presented in this study are included in the article/Appendix A. Further inquiries can be directed to the corresponding authors.

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
