# Peer review of "Study of Reaction Parameters for the Precise Synthesis of Low-Molecular-Weight Oligosiloxanes"

_materials, 2025, doi:10.3390/ma18245677_

Round 1

Reviewer 1 Report

Comments and Suggestions for Authors

The manuscript presents a systematic experimental study on the synthesis of low-molecular-weight oligosiloxanes (DP < 15) by anionic ring-opening polymerization of cyclic trisiloxanes (D₃). The authors investigate how the reaction parameters — initiator type (MeLi vs n-BuLi), promoter-to-initiator ratio, monomer-to-initiator ratio, reaction time, and solvent composition — affect the degree of polymerization. The work is clearly written and the experiments appear carefully executed. Although the results are largely empirical and mechanistic insight remains limited, the study may have practical value for laboratories and industries interested in the controlled preparation of siloxane oligomers, a field with few recent quantitative reports.

However, several points require clarification before publication.
(1) The rationale proposed for the different behavior of MeLi and n-BuLi initiators is unconvincing: once silanolate formation occurs, the alkyl group no longer remains bound to the polymer chain. The differences observed most likely arise from the distinct solvents used (Etâ‚‚O for MeLi, hexane for n-BuLi) and the resulting solvation of the lithium cation.
(2) In Table 4, the volumes of hexane, toluene, and THF are changed independently, so the total reaction volume and initiator concentration are not constant. Because anionic ROP is concentration-sensitive, the apparent solvent-polarity effects may partly reflect dilution.
(3) The reported clean isolation of dimethylvinylsilanol after aqueous work-up and drying at 60 °C is chemically doubtful; such small silanols typically condense rapidly to disiloxanes. Analytical evidence or a clarification on whether the compound was used in situ should be provided.

Overall, the study is competently executed and potentially useful as a technical contribution. Provided the authors address the above issues and refine their discussion accordingly, the manuscript could be accepted after minor revision.

Author Response

We sincerely appreciate your time in reviewing our manuscript and your valuable comments.

Our responses are provided in the attached PDF file.

In addition, in response to the editorial broad comments, we revised the sentences on pages 7 and 8 (pages 2 and 3 in the revised version) in the experimental section, relocated the Materials and Methods section before the Results and Discussion section, and added seven new references (now listed as References 16-19, 23, 24, and 26 in the revised manuscript).

Reviewer 2 Report

Comments and Suggestions for Authors

The reviewed paper is aimed at the carefully contgrolled synthesis of oligosiloxanes with low degrees of polymerization (<15). The authors thoroughly investigated the effect of the ratio of the components on the molecular weight and polymerization  degree of the formed  siloxanes, and found optimal conditions allowing to control the molecular weights by varying the ratio of the promoter to initiator, as well as the solvent. 

The paper can be slightly improved by polishing the English language or presentation (e.g., Fig. 2 could be omitted) but, in principle, these minor corrections can be made on the stage of checking the proofs, so, I would recommend to accept the paper as it is, in spite of the absence of outstanding scientific novelty.

Author Response

We sincerely appreciate your positive and encouraging feedback. We have revised portions of the English, particularly in the experimental sections, and we will further refine the presentation during the proof-checking stage.

In addition, in response to the editorial broad comments, we revised the sentences on pages 7 and 8 (pages 2 and 3 in the revised version) in the experimental section, relocated the Materials and Methods section before the Results and Discussion section, and added seven new references (now listed as References 16-19, 23, 24, and 26 in the revised manuscript).

Thank you again for your constructive comments and recommendation for acceptance.

Reviewer 3 Report

Comments and Suggestions for Authors

The authors investigated influence of various parameters on the synthesis of oligosiloxanes with degrees of polymerization below 15. The methods for synthesizing oligosiloxanes with precisely controlled molecular weight and degrees of polymerization are demonstrated. Low-molecular-weight polysiloxanes with well-defined molecular characteristics could attract attention due to their versatile functional properties and potential applications. Although some studies have explored the control of polysiloxane molecular weights, precise regulation of oligosiloxane molecular weight has been rarely investigated. This study aims to establish optimized reaction conditions for the synthesis of oligosiloxanes with precisely controlled molecular weights. The results revealed that molecular weight of oligosiloxanes can be effectively tuned by adjusting the molar ratio between the promoter and initiator, the initiator and cyclotrisiloxane (D3), as well as by varying the lithium type and solvent composition in the ring-opening polymerization. The results are interesting and the paper could be accepted after revision.

-The authors should demonstrate potential of the prepared oligosiloxanes in applications as of advanced materials.

-Synthesis: The synthesis of oligosiloxanes was performed under an argon atmosphere using standard Schlenk techniques. Tetrahydrofuran (THF) and toluene were dried using solvent purification system. The synthesis should be complicated and expensive for the conditions.  Is the method useful economically for the practical application of the oligomers ?

-The authors declare in conclusions that potential potential applications of the oligomers could be as precision materials and functional monomers.  This should be demonstrated by practical experiments.

-GPC curves should be demonstrated for the oligomers.

-According to MS spectra there are some low molecular weight materials in the oligomers ?  How they could be removed from the oligomers?

Yields of the reactions should be described.

Author Response

We sincerely appreciate your time in reviewing our manuscript and your valuable comments.

Our responses are provided in the attached PDF file.

Round 2

Reviewer 3 Report

Comments and Suggestions for Authors

Yields of the reactions should be described. It is not clear if the synthesis is perspective without the information about yield.

Author Response

Yields of the reactions should be described. It is not clear if the synthesis is perspective without the information about yield.

Response: Thank you very much for taking the time to revise our manuscript, and we truly appreciate your insightful comment. We agree that the reaction yields are essential for assessing the practicality of the synthetic procedures. Accordingly, we have added the yields for all relevant experiments below Tables 2-4 of the revised manuscript. These additions have been highlighted in yellow.